# Digital Financial Inclusion and Remittances: An Empirical Study on Bangladeshi Migrant Households

**Kazi Abdul Mannan** [1,*] and **Khandaker Mursheda Farhana** [2]

1 Faculty of Business Studies, Green University of Bangladesh, Kanchon 1460, Bangladesh
2 Department of Sociology and Anthropology, Shanto-Mariam University of Creative Technology, Dhaka 1230, Bangladesh; drfarhanamannan@gmail.com or drfarhana@smuct.ac.bd
* Correspondence: drkaziabdulmannan@gmail.com or drmannankazi@capdr.org

**Abstract:** Globally, large numbers of adults remain unbanked, and most of them live in rural areas of the Third World. The recent outbreak of the COVID-19 pandemic has shown us how inequalities in accessing financial services continue to affect us. However, digital financial inclusion has emerged as an effective tool used to tackle socioeconomic ills and drive economic development. In fact, due to these modern technological developments, the number of studies in this area is very limited, especially in the context of developing economies. This study examines the impacts of migrant remittances on digital financial inclusion within households in Bangladesh by using the Migration and Remittance Household Survey. To meet the research objectives of this study, a household survey was conducted and 2165 households interviewed in 2022–2023 in Bangladesh. The survey data collected was tested using univariate and multivariate estimations. This study finds that the coefficient of remittance has positive relationships with the probability of e-bank accounts and the use of mobile banking for a household's financial transactions. However, the use of ATM cards by households for financial transactions has not been significantly affected. The article concludes that remittance flows may enhance access to and use of means of digital financial inclusion by reducing some of the barriers and costs in Bangladesh, which could greatly contribute to the country's economic growth by creating and increasing a strong fund for investment. The findings of this study can help in taking various steps to facilitate the most powerful financial sector of Bangladesh, namely, remittance management.

**Keywords:** digital financial inclusion; migration; remittance; household; rural–urban; Bangladesh

## 1. Introduction to and Background of the Study

The results of ongoing technological innovation are spreading their influence in every sector of society, and, hence, inclusion in the financial sector also contributes to inclusive socio-economic development. Economic growth fueled by digital technologies attracts significant attention in academic discourse in business and economic research [1–3]. Recently, emerging digital financial services have attracted the attention of stakeholders, especially policymakers and scholars, as a potential gateway for digital financial inclusion. The use of digital payment technology, Internet-enabled money-transfer systems, and mobile phone technology through digital platforms has made financial systems more accessible. It can be seen that, in many developing countries, financial inclusion appears to be a potential transformative agent that can reduce poverty and contribute to creating more financially inclusive societies [2]. Recently, the political and economic slogan in Bangladesh has been to build a Digital Bangladesh. However, in the era of cyber technology, no institutional definition has been made for building a digital Bangladesh. However, what can be understood is that the main goal is to make the country's financial sector rapidly develop and pioneer various inclusions using modern technology. A large population of Bangladesh is still immersed in the darkness of illiteracy, but the use of mobile phones has become a very easy matter even for this illiterate population.

Despite the significant increase in financial-sector inclusion, little attention has been paid to understanding the potential determinants of digital financial-inclusion issues and the challenges of digital approaches. While the concept of digital financial inclusion has been explored in the existing literature, the digital divide and underlying aspects of social inclusion in access to digital finance remain extremely veiled [4]. However, recent studies have explored the impact of digital financial inclusion and mobile financial services in Bangladesh, highlighting how digital financial services constitute a new business phenomenon and market-entry strategy in Bangladesh [5]. Moreover, based on the information and communication technology (ICT) policy of Bangladesh, a study found that the concept of digital inclusion was not strongly reflected in governmental digital projects and policy strategies [6]. So, it remains unknown how much digital financial technology and digital services ensure digital financial inclusion, especially in terms of remittances, currently the strongest financial sector in Bangladesh.

The basic objective of this study is to find the effect of migrant household remittances on digital financial services in Bangladesh. Specifically, the aim is to observe what induces the use of digital financial inclusions among Bangladeshi remittance-recipient households. The outcomes also provide enlightenment as to whether an increase in migrant household remittances will increase the digital financial product position of Bangladeshi households.

The first mobile banking service was launched in Bangladesh in 2010. Currently, there are 15 mobile banking services in Bangladesh. At present, development services are available in all parts of the country, in cities, towns, villages, and towns. The cash mobile banking service was launched in 2019. Under mobile banking services, banks appoint banking agents to perform banking activities on their behalf, such as opening mobile banking accounts and providing cash services (receipts and payments). Cash withdrawals from a mobile account can also be made from an ATM that validates each transaction by 'mobile phone and PIN' instead of 'card and PIN'. Other services provided through the mobile banking system are person-to-person (money transfer), person-to-business (commercial payments and utility bill payments), business-to-person (salary/commission disbursement), and government-to-person (Disbursement of Government Allowance) transactions.

Currently, remittances can be sent directly from 65 countries of the world through mobile banking. Migrants who want to send money to their family from abroad can easily send money to their family's mobile number. In this respect, Bikash is a popular mobile banking service in Bangladesh. Many times, expatriates have been cheated while trying to send their hard-earned money to their families through others. In this case, if an expatriate sends money from abroad directly to their family's mobile number, there is no opportunity for anyone else to embezzle that money. Expatriate Bangladeshis living abroad can easily and conveniently send money to their loved ones' mobile accounts in Bangladesh through authorized and listed foreign banks, money transfer organizations, and money-exchange houses.

The recent COVID-19 pandemic has reduced global migration flows by 27% [7]. In addition to this, several other factors, such as increased health concerns, job losses, and travel restrictions, have also had considerable negative impacts on many migrant workers who send remittances to their families in their home countries. A report shows that in 2020, the average flow in the sector for countries receiving global remittances declined by 1.5%, to a total of USD 711 billion [8]. However, over the next two years, the pandemic coincided with some easing of the bottlenecks, i.e., as international borders opened and visa approvals resumed, the international migration system regained some normal momentum, resulting in increased global remittance flows. According to World Bank data, in 2021, total global remittances were estimated at USD 781 billion, and further increased to USD 794 billion in 2022 [9]. However, the growing uncertainty and elevated inflation worldwide adversely affected migrants' real income and their remittances. Bangladesh's remittance earnings for financial year 2022 stood at USD 21,031.68 million, and the remittance–GDP ratio, remittance–export earnings ratio, and remittance–import payments ratio were 4.56% percent, 42.71% percent, and 25.49% percent, respectively, in financial year 2022 [10].

The motivations for sending remittances are mainly composed of a summation of socioeconomic reasons, such as altruism, investment, self-interest, debt repayment, and social-commitment purposes [11,12]. Moreover, an ample body of literature has arisen on the impact of household remittances on poverty, inequality, investment in small enterprises, resource mobilization, health, and education [13–19]. However, studies of the impact of remittances on digital financial inclusion from the perspective of Bangladesh are rare. In this study, using a survey of remittance-receiving households in Bangladesh, a hypothesis is tested: that remittances affect the use of digital financial services.

Evidence from previous studies shows that migrant household remittances may affect different types of digital financial inclusion through multiple pathways. Firstly, sending remittances by official channels can raise household needs, as bank savings levels affect levels of government support. Since different fixed costs and variable expenses are associated with sending remittances, migrants may also send remittances to families irregularly, which can save a family extra cash, but only for a limited period. Remittances may also raise households' requirements for savings accounts in which to keep the money, given the informal nature of remittances [20–23]. Secondly, if migrants send remittances through formal channels, their families can be informed by financial institutions about the institutions' various bank-loan products. Moreover, since financial institutions keep important information about remittance-receiving households, the trust of financial institutions in extending various types of loans to these households can be increased [21,24]. Thirdly, financial institutions, through the accumulation of remittances, not only help meet the credit needs of remittance-receiving households, but also extend the same utility to other households and organizations [25–27].

To solve the research problems, this paper applies an estimation of instrumental variables (IV) and propensity score matching (PSM) analysis. Therefore, this study uses migrant network effects to control for expected endogenous effects of migrant household remittances to construct an impartial and coherent analysis of the effect of remittances on financial services. There may be endogenous effects of migrant remittances received by households, and this necessitates the use of an instrumental variable technique that we know to be generally reliable for determining the probabilities of causal relationships and reverse causality. In addition, the PSM method was adopted as a robustness test to verify the results derived from the study's hypothesis.

The outcomes of this study establish that households receiving remittances in Bangladesh have an increased probability of using savings accounts and adopting mobile banking. This study closely relates to an increasing body of literature that conceptualizes the impact of migrant household remittances on financial products, and it provides extensive context for the literature that explains the impact of migrant household remittances on the financial area and on national economic development [14,16,17]. Nevertheless, this paper distinguishes itself from the existing literature, as its findings are determined from the perspective of a developing country, specifically, Bangladesh. It is known that Bangladesh was the seventh-highest recipient of remittances in the world, at almost USD 22.1 billion, in 2021, when it was also the third-highest recipient of remittances in South Asia [28]. This project contributes a distinctive and captivating study of the region, as well as empirical research.

The contemporary relevance of this paper looks to the benefits of substantive financial services, which have been emphasized in the literature. Empirical evidence shows that remittances increase consumption, income, employment, education, access to microfinance, and mental health [29–31]. Moreover, incoming remittances for microfinance may assist in enormous financing opportunities in commercially viable products, increase the number of start-up businesses, and improve the profitability of existing enterprises [32]. Additionally, household savings have been shown to increase access to saving [33,34], investment [35], consumption [35,36], and women's empowerment. Eventually, digital financial inclusion is significantly correlated with household economic development, which may provide a route to investment and economic growth in the development of the state [37–39].

## 2. The Theoretical Background of the Study

Before starting research, it is very important to check basic studies of the past related to the topic, a tactic which guides the research in the right direction. Considering that aspect, this study logically supports the technology acceptance model (TAM), mobile banking acceptance model (MBAM), and mobile banking resistance model (MBRM). It can be seen that research methods differ depending on the research problem and objective. Contemporary research on mobile banking and digital financial inclusion has led this research [40]. One breakthrough in this new paradigm is that even those who do not have an e-bank account with a financial institution can avail themselves of this service. These services include depositing and withdrawing salaries, paying various utility bills, receiving domestic and international remittances, receiving and repaying loans, processing airtime purchases, and making various purchases [41].

We see that, along with technological innovations, some new models are being used in various research in connection with TAM, such as diffusion of innovation, unified theory, and trust-based models [42] Coincidentally, the beginning of research on financial technology and technology acceptance coincides largely with TAM. However, the newly developed theories aim to determine the extent to which the acceptance factors of emerging technologies influence the level of acceptance at the individual level and how they are related to the emerging technologies. The TAM model has been used in a single, institutional way, especially for the development of an organization, to see what kind of technology use can help the overall development of the establishment [42]. That is, this model is an idea that explains how people will use a new technology.

However, various analyses by psychologists show that a better understanding of human behavior is best achieved with cognitive theory, which is often expanded to include the theory of planned behavior [43]. Fishbein and Ajzen conceived the theory that is now used individually and collectively to describe differences in human behavior and attitudes. There are some ways in which the concept of rational action can be separated from information-integration theory. As such, attention to one's behavior may limit the influences of perceptions and attitudes on behavior. The technology readiness and acceptance model, also known as TRAM, is an amalgamation of the two separate models, as the name suggests, by Fishbein and Ajzen. The TRAM is the most recent contribution to the combining of TAM's dimensions to further specify personality traits of human character [6]. Finally, this article will describe how different personality factors can influence how family members engage with, experience, and use new technology [44].

## 3. Literature Review

Recent studies have examined the impact of remittances on different household activities, for instance, entrepreneurship [45–48], poverty and inequality [16,49], and education and wellness [14]. Evidence from developing nations shows that remittance-receiving households are generally more motivated to invest in the housing sector [46,50]. Moreover, it also shows that remittance receipt has a positive impact on employment, productivity, and micro-macroeconomic development. However, some also argue that remittances can result in a discouraged labor supply, which creates a cycle of financial dependence by reducing recipients' motivation to work [51]. Furthermore, in most countries, remittance-recipient households tend to spend money on various types of luxury goods rather than invest in physical assets [52–54].

In previous studies, research on remittances and digital financial inclusion has received much attention, mainly among policy-development experts. This increasing interest in digital financial inclusion is a result of the day-by-day drive towards digital financial inclusion in households, societies, and financial development [33–35,54]. Hence, in this background survey, a rising body of literature is building toward understanding the implications and impacts of household remittance receipts and new financial inclusion [20–23].

The existing literature thus far provides us with two main perspectives on the association between migrant household remittances and digital financial inclusion. Firstly,

remittances are a simple credit alternative. This ramification arises from a theoretical and analytical framework in which an imperfect credit market exists, and where remittances assist marginal and liquidity-strapped households' investment in physical or human capital and alleviate the effect of shocks by financial crises [55–58]. Secondly, there is an increasing body of evidence that remittances have a significant impact on savings, at both the macro- and micro-economic levels [20,22,59,60]. A few of the causes of the positive effects of remittances include the weighting of remittances in the savings index, which can essentially create demand for remittance recipients in deposit accounts; expanding knowledge about digital financial inclusion products; the matching of information from other users; and above all, the enhancement of some of the participants' financial opportunities and creditworthiness by receiving remittances [23,26].

From the above review, it is clear that there are limitations to the literature on digital financial inclusion related to remittances and socioeconomic outcomes in households, especially with respect to developing countries. It is therefore hoped that this study will be able to contribute to the existing literature in this field, which will help inform policy development and discussions.

## 4. Conceptual Framework and Hypothesis Development

Information communication technology (ICT) continues to grow at an unprecedented rate and disrupts technological use; researchers are interested in the factors influencing users' acceptance of a particular technology, including their readiness to use it. To address these technological issues, researchers are constantly developing new methods and techniques to measure the acceptance of new technologies, while also examining how these technologies impact society. A recently developed 36-item scale named the technology readiness index measures the acceptance, use, and impact of new technologies in both family and work environments [61]. TAM is an advanced model that easily integrates other models of technology. The integration of multiple theories can be seen as the reason [62] that TAM and TRI together can explain the adoption of new technologies [63]. Moreover, individual differences are mediated by cognitive dimensions in predicting people's acceptance of new technologies [64]. A graphical representation of the proposed research model and hypothesis is depicted in Figure 1.

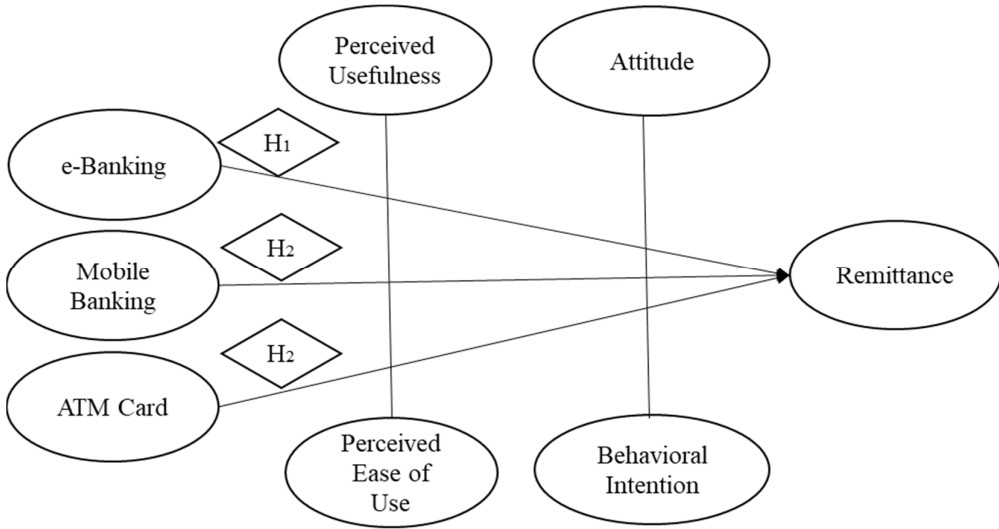

**Figure 1.** Conceptual Model. Source: the authors; developed for the present study.

Based on the above discussion, three hypotheses have been formulated for this study, which are:

**H1.** *There is a relationship between use of e-banking and being a remittance-recipient's household.*

**H2.** *There is an association between using mobile banking and being a remittance recipient's household.*

**H3.** *The attitude towards an ATM Card affects the behavioral intention to use it.*

In this study, e-banking is defined as a system in which all types of transactions can be performed online. Mobile banking refers to accomplishing all types of financial transactions through just a mobile number, rather than conventional banking means. The ATM card method refers to the deposit and withdrawal of cash through a card, using the traditional banking system. In Bangladesh, these three systems together are considered by the state to be digital financial inclusion in the financial sector.

## 5. Methods of the Study

This paper uses a household survey that was conducted by means of a stratified random sampling method that included 16 districts out of the 8 divisions at the first-order administrative level (consisting of 64 districts) and the capital city in Bangladesh, and involved interviews with 2165 households, which were conducted in 2022–2023. The household survey was nationally representative and involved a one-time collection of data on migration and remittances from one migrant household. In terms of sample selection, using the migration data of the Manpower Employment and Training Bureau of the Government of Bangladesh, 16 districts were selected from among the districts with the highest number of migrants within the 8 divisions. Every district headquarters has an information repository for migrant families. One district is considered to be a stratum in this study. A stratified sub-group sample was determined in proportion to the population of each district from the total sample determined.

In this study, we have used the SERVQUAL instrument (incorporating reliability, responsiveness, assurance, empathy, and tangibles) to develop the questionnaire [65]. To collect this data, a self-administered questionnaire was developed that focused on themes related to participants' experiences of accessing and using digital technologies for financial services. Prior to data collection, the field workers were properly trained, with particular care being taken relative to the ethical issues of obtaining consent from each participant and maintaining confidentiality. In this case, eight research associates and eight research assistants, one pair in each of the eight divisions, completed all the questionnaire surveys in three months under the direct supervision of the principal researcher. Among the important data, there were mainly preferences for results relevant to family digital financial inclusion. For example, it was important to determine whether the household had its own savings account, the extent to which the household used mobile banking, and the extent to which ATM cards were used for financial transactions. Indeed, the focus of this article is to examine the impact of household remittance receipts on access to digital financial inclusion.

The Statistical Package for Social Science (SPSS) software (https://www.ibm.com/products/spss-statistics, accessed on 15 October 2023) was used for analyzing and interpreting the data. The sources were coded for the analysis of data. In this case, two types of coding were used: alphabetic coding for the district, and alphanumeric coding for participants. The reliability of the responses and the Cronbach alpha values for each construct were checked. The results of all the constructs demonstrate high internal consistency, ranging from 0.60 to 0.86, which is greater than the threshold value of 0.60 [66]. Descriptive statistics were used to compile the data. The means and standard deviations of the other dependent and independent variables, as well as the mean and standard deviation of the primary variables, were calculated.

This paper analyzes the relationship between household remittance receipt and the use of digital financial inclusion, using the following model:

$$FinI_h = \beta + \alpha_1 \, Rm_h + \alpha_2 \, C_h + \varepsilon_h \tag{1}$$

where:

h = the household;

FinI = 1, if uses of digital financial inclusion, and otherwise = 0;

Rm = households' covariates;

$\varepsilon$ = the error term.

In this study, we followed [20] by determining the controlling variables, which were the age of the household head; the gender of the household head; the household head's level of education; the household's size, number of dependents, and number of female members; the destination of the migrants; the activity of the migrants; and a regional variable as a dummy. An empirical analysis estimated Equation (1) by using a linear probability model. This study also described the marginal impacts of probit regressions for the diverse financial services determinatives.

Using two models such as the linear probability model and the probit model can raise an important problem, namely, that the hypotheses that were proved may have been endogenous due to the omitted variables. Research estimates may be biased in cases where omitted variables are related to households' likelihood of receiving remittances and using digital financial inclusion. Therefore, this problem is best solved by using an instrumental variable analysis. Moreover, an opposite causation may influence these analyses from Equation (1), because access to digital financial inclusions may raise the issue of the simplicity of sending and receiving remittances. Thus, this consideration probably increases the probability of migrants' remittance inflow.

This study adopts an instrumental variable, because this tactic can eliminate potential bias arising from the possible endogeneity of the results. In the migration and remittances literature, these instruments are mainly called migrant network impacts [13,20,67]. A migrant's destination or network may influence the probability of migration and the receipt of remittances by their family, both of which may serve as instruments for the migrant network effect. Indeed, it is not expected that migrant network effects will influence households' access to and use of digital financial inclusion.

The reliability and validity of the identification strategy used in this article depend on the instrument's (the network effect of migrants) satisfaction of both assumptions. Firstly, migrants' remittances are positively correlated with the instrumental variables analysis. Secondly, those factors that may always affect household digital financial inclusion are assumed, namely those that are most likely to be correlated with the instruments. Additionally, it is assumed that the instrument meets exclusion restrictions. From this, it is clear that the instrument does not affect the direct-effect outcome, financial service, without the first-level regression.

Robustness tests are performed using population propensity score matching (PSM) to set forth potential selection bias connected to the selection of unobservable characteristics that might create a relationship between the propensity to receive remittances and the likelihood of using financial services. Migration and remittance studies show that the PSM approach allows for potential selection corrections by differentiating between remittance-receiving households and non-remittance-receiving households depending on their propensity score [68,69].

In the propensity score matching approach, the status of digital financial inclusion is defined using households not receiving remittances as the control group and households receiving remittances as the treatment group, one which exhibits opposite results. Various studies have shown that PSM provides more accurate non-experimental estimates when households self-select into a treatment group, compared to other estimation methods [70]. Below are the equations, specifically:

Let

$R_i$ = 1 (if received remittances)

$R_i$ = 0 if not received remittances

$F_{1i}$ = digital financial Inclusion—a household receiving remittances

$F_{0i}$ = digital financial inclusion-a household non-receiving remittances

$$\delta F_i = K (F_{1i} | R_i = 1) - K (F_{0i} | R_i = 1) \tag{2}$$

Above, Equation (2) shows that it is impossible to observe two distinct types of families at the same time. Therefore, the result of household remittance receipts could be examined, but the same result may not be observed in that case as would be seen in cases with the non-existence of a remittance receipt. The propensity score matching estimations depend on the unpredictable independence of acceptance, which explains, conditional on Z, that the dynamic effects are independent of the treatment positions (receiving remittances). The treatment assignment is as proper as a random variable, since it controls for the observable covariates, Z [70].

Applying propensity score matching, the parameter of interest in this study is the average treatment effect on treatment, which is calculated by subtracting the average treatment effect of the treated group from the control for a given propensity score.

Thus,

An average treatment effect on treatment (ATET) = K [F | Y = 1, L(Z)] − K [F | R = 0, L (Z)]

## 6. Findings

Migrants have migrated to different cities within the country and also to places outside the country. As shown in Figure 2, about 57% of the migrants migrated within Bangladesh (mainly from rural areas to urban areas), while, among the other countries, 10% migrated to various European countries, 22% to the Middle East, 9% to East Asia, and 3% to other destinations. Among total internal migrants, 41% were employed in the government and private sector, 22% were self-employed, 20% were students, and the remainder were not engaged in any specific labor force or were engaged in other activities, as shown in Figure 3. The majority of migrants in European countries were involved in paid employment (63%), and others were engaged in small-scale trade. Most (97%) in the Middle East, as well as in other countries, were engaged in paid employment, but there were also some students among those in other countries.

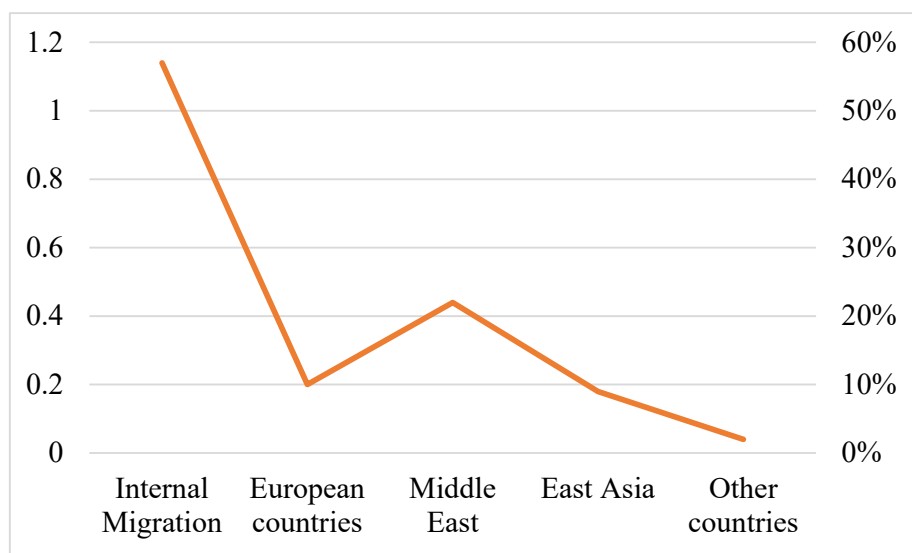

**Figure 2.** Destination of migrants. (Authors' computation.).

Table 1 presents descriptive statistics for household characteristics. The average annual remittance of each family was BDT 475,000, of which 87% were from international remittances and 13% were from internal remittances. A total of 63% of households held an e-bank account, 21% used mobile banking, and only 7% used ATM cards for cash transactions. There were no significant numerical differences between rural and urban

participants. The average adult had a little more than nine years of formal education, but the average head of household had a little more than ten years of formal education. The average household size was about four people and the average age of the household head was 36 years. Households headed by a female constituted 37.8 percent of total households, while the proportion of female members in a household was 32.5%. The family dependency ratio can be seen to be at 43.4%.

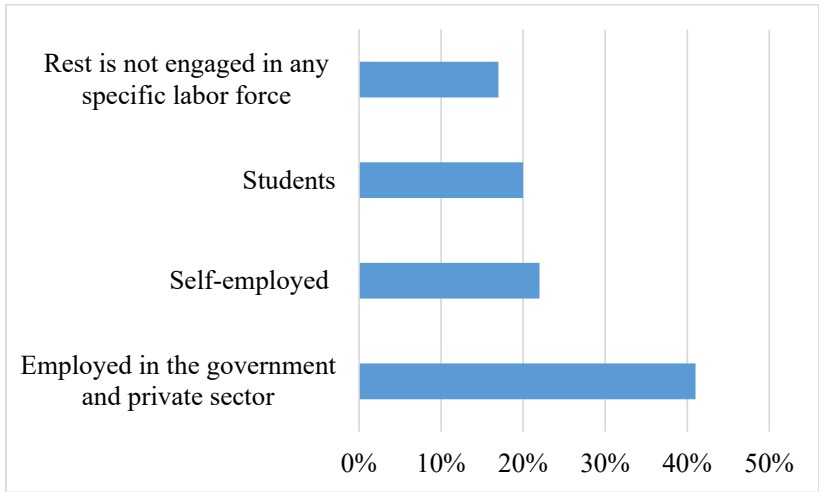

**Figure 3.** Employment status of internal migrants. (Authors' computation.).

**Table 1.** Descriptive Statistics.

| Variable | Mean | Std. Dev. |
|---|---|---|
| Remittance (Internal) | 163.211 | 628.332 |
| Remittance (International) | 144.334 | 598.243 |
| Household has an e-bank account | 0.635 | 0.534 |
| Household uses mobile banking | 0.215 | 0.521 |
| Household uses an ATM card | 0.072 | 0.543 |
| Household head: level of education | 10.324 | 5.361 |
| Number of household members | 4.057 | 3.145 |
| The average age of the household members | 36.325 | 8.822 |
| Dependents (%) | 43.715 | 21.825 |
| Female member of the household (%) | 37.567 | 18.354 |
| Region (Urban) | 0.497 | 0.499 |
| Gender (Female migrant) | 0.169 | 0.457 |

Table 2 presents the mean differences that represent the characteristics of households surveyed, whether remittance had been received or not. In this paper, digital financial inclusion is proxied by three indicators: the household having an e-bank account, the household using mobile for financial transactions, and the household using ATM cards for financial needs. Analysis of the results indicates that the majority of households that received remittances had at least one e-bank account. Moreover, about 21.8 percent of remittances were received by households that used mobile banking for financial needs, and only 9.3 percent of recipient families used ATM cards for financial needs. It is noted here that most of the women-headed families that received remittances lived in rural villages.

Probit regression results are presented in Table 3 below, and the disaggregated (internal and international remittances) analyses are presented in Tables 4 and 5, respectively. Both Tables 4 and 5 use linear probabilistic regression estimation techniques. Each table shows, in different columns, the effect of migrant remittances against the probability that the household has an e-bank account, that the household uses mobile banking, and that the household members use ATM cards for their financial transactions.

**Table 2.** Mean Difference (*t*-test).

| Variable | Received Remittance | | Difference |
|---|---|---|---|
| | **No** | **Yes** | |
| Remittance (Internal) | | | |
| Remittance (International) | | | |
| Household has an e-bank account | 0.621 | 0.721 | |
| | (0.415) | (0.422) | −0.047 ** |
| Household uses mobile banking | 0.315 | 0.288 | |
| | (0.427) | (0.417) | 0.071 ** |
| Household uses an ATM card | 0.074 | 0.064 | |
| | (0.409) | (0.438) | 0.077 ** |
| Household head: level of education | 10.334 | 9.385 | |
| | (4.147) | (4.333) | 0.217 |
| Number of household members | 4.879 | 4.709 | |
| | (1.726) | (2.357) | −0.411 *** |
| The average age of the household members | 36.384 | 39.821 | |
| | (8.373) | (9.217) | −1.784 *** |
| Dependents (%) | 34.331 | 29.473 | |
| | (23.928) | (23.548) | 3.245 *** |
| Female member of the household (%) | 31.154 | 33.825 | |
| | (15.377) | (16.342) | −2.316 ** |
| Region (Urban) | 0.498 | 0.572 | |
| | (0.428) | (0.433) | −0.137 *** |
| Gender (Female migrant) | 0.087 | 0.549 | |
| | (0.208) | (0.477) | −0.077 *** |

Note: ** and *** represent significance levels at 5% and 1%, respectively.

**Table 3.** Total remittance: probit regression results.

| | Total Remittance | | |
|---|---|---|---|
| | **The Household Has an e-Bank Account** | **Household Uses Mobile Banking** | **Household Uses an ATM Card** |
| Remittance (Log) | 0.007 *** (0.001) | 0.003 (0.002) | 0.002 (0.002) |
| The age of the household head | 0.035 *** (0.002) | 0.015 *** (0.004) | 0.026 *** (0.003) |
| Number of household members | 0.028 *** (0.007) | 0.006 (0.008) | 0.004 (0.007) |
| The household head's level of education | 0 (0.002) | −0.006 *** (0.001) | −0.004 *** (0.001) |
| Female household head (%) | 0 (0.002) | 0(0.002) | −0.001 **(0.002) |
| Destination: | | | |
| European countries | 0.023 (0.033) | 0.007 (0.022) | 0.061 ** (0.022) |
| Middle East | 0.006 (0.034) | 0.221 ** (0.044) | 0.062 (0.048) |
| East Asia | −0.022 (0.061) | −0.082 (0.067) | −0.025 (0.065) |
| Other countries | 0.051 (0.062) | 0.043 (0.202) | 0.006 (0.073) |
| Employment status of migrants: | | | |
| Paid employment | −0.050 * (0.018) | −0.022 (0.032) | −0.035 (0.037) |
| Small-scale trade | 0.002 (0.017) | 0.006 (0.044) | −0.030 (0.280) |
| Student | −0.02 (0.072) | 0.232 (0.085) | −0.024 (0.078) |
| Unemployment | 0 (0.04) | 0.013 (0.20) | 0.020 (0.072) |
| Others | −0.022 (0.072) | 0.043 (0.033) | −0.008 (0.077) |
| Gender (Female migrant) | −0.075 ** (0.24) | −0.46 (0.034) | −0.071 * (0.030) |
| Const. | 0.028 (0.084) | 0.477 (0.243) | 0.424 *** (0.221) |
| Rural | Yes | Yes | Yes |
| Region | Yes | Yes | Yes |
| R-squared | 0.411 | 0.156 | 0.283 |

Note: *, **, and *** represent significance levels at 10%, 5%, and 1%, respectively.

**Table 4.** Internal remittance: probit regression results.

| | Remittance (Internal) | | |
|---|---|---|---|
| | **The Household Has an e-Bank Account** | **Household Uses Mobile Banking** | **Household Uses an ATM Card** |
| Remittance (Log) | 0.007 *** (0.003) | 0.003 (0.002) | 0.002 (0.002) |
| The age of the household head | 0.034 *** (0.002) | 0.031 *** (0.004) | 0.016 *** (0.003) |
| Number of household members | 0.028 *** (0.007) | 0.006 (0.008) | 0.004 (0.007) |
| The household head's level of education | 0.0 (0.002) | −0.006 *** (0.003) | −0.003 *** (0.002) |
| Female household head (%) | 0.0 (0.002) | −0.002 (0.002) | −0.002 (0.001) |
| Destination: | | | |
| European countries | 0.017 (0.033) | 0.008 (0.022) | 0.062 (0.042) |
| Middle East | 0.02 (0.034) | 0.211 ** (0.049) | 0.063 (0.048) |
| East Asia | −0.022 (0.042) | −0.081 (0.068) | −0.029 (0.088) |
| Other countries | 0.043 (0.061) | 0.042 (0.011) | 0.007 (0.073) |
| Employment status of migrants: | | | |
| Paid employment | −0.044 * (0.018) | −0.011 (0.032) | −0.037 (0.029) |
| Small-scale trade | 0.003 (0.019) | 0.007 (0.028) | −0.028 (0.041) |
| Student | −0.020 (0.072) | 0.021 (0.083) | −0.023 (0.078) |
| Unemployment | 0.0 (0.040) | −0.072 (0.075) | 0.010 (0.071) |
| Others | −0.024 (0.071) | 0.019 (0.101) | −0.008 (0.070) |
| Gender (Female migrant) | −0.072 ** (0.024) | −0.048 (0.039) | −0.072 * (0.030) |
| Const. | 0.041 (0.081) | 0.475 *** (0.135) | 0.049 *** (0.133) |
| Rural | Yes | Yes | Yes |
| Region | Yes | Yes | Yes |
| R-squared | 0.411 | 0.204 | 0.283 |

Note: *, **, and *** represent significance levels at 10%, 5%, and 1%, respectively.

**Table 5.** International remittance: probit regression results.

| | Remittance (International) | | |
|---|---|---|---|
| | **The Household Has an e-Bank Account** | **Household Uses Mobile Banking** | **Household Uses an ATM Card** |
| Remittance (Log) | 0.007 *** (0.003) | 0.003 (0.004) | 0.002 (0.004) |
| The age of the household head | 0.039 *** (0.004) | 0.034 *** (0.004) | 0.026 *** (0.003) |
| Number of household members | 0.030 *** (0.007) | 0.007 (0.008) | 0.004 (0.007) |
| The household head's level of education | 0.0 (0.001) | −0.007 *** (0.003) | −0.004 *** (0.003) |
| Female household head (%) | 0.0 (0.001) | −0.002 * (0.001) | −0.001 ** (0.002) |
| Destination: | | | |
| European countries | −0.006 (0.037) | 0.004 (0.048) | 0.076 * (0.043) |
| Middle East | −0.020 (0.031) | −0.107 ** (0.049) | 0.077 (0.072) |
| East Asia | −0.041 (0.041) | −0.088 (0.068) | −0.039 (0.071) |
| Other countries | 0.021 (0.064) | 0.033 (0.216) | 0.001 (0.076) |
| Employment status of migrants: | | | |
| Paid employment | −0.074 ** (0.031) | −0.017 (0.030) | −0.040 (0.027) |
| Small-scale trade | −0.020 (0.015) | −0.011 (0.027) | −0.044 (0.029) |
| Student | −0.048 (0.071) | 0.121 (0.084) | −0.039 (0.076) |
| Unemployment | −0.031 (0.038) | −0.069 (0.074) | 0.001 (0.073) |
| Others | −0.071 (0.074) | 0.02 (0.087) | −0.031 (0.079) |
| Gender (Female migrant) | −0.063 ** (0.41) | −0.49 (0.039) | −0.73 (0.028) |
| Const. | 0.082 (0.082) | 0.061 *** (0.201) | 0.49 *** (0.211) |
| Rural | Yes | Yes | Yes |
| Region | Yes | Yes | Yes |
| R-squared | 0.385 | 0.155 | 0.286 |

Note: *, **, and *** represent significance levels at 10%, 5%, and 1%, respectively.

The probit regression results indicate that a positive impact of remittances was seen on all digital financial inclusion indicators for households, but also that their coefficients were significant, that is, in determining if a family member held an e-bank account. These

results primarily support the first hypothesis (H1). This analysis obtains regardless of total remittances (Table 3). Additionally, Tables 4 and 5 explore the proposition that having at least one e-bank account in the household receiving remittances increases the likelihood of digital financial inclusion, in other words, a unit change in migrant remittances increases the probability of household members holding an e-bank account. This outcome is certainly plausible given that having an e-bank account can reduce the expenditure of remittance transfers, thereby increasing migrant remittance flows.

Table 6 shows the regression results, which indicate the impact of total remittances, in light of instrumental variables, on digital financial inclusion in Bangladesh. However, Tables 7 and 8 report the analyses of separately-collected data on internal and international remittances, respectively, on digital financial inclusion. The two-stage least squares regression estimates come together with the results of the linear probability model and probit estimates. The two-stage least squares regression is more robust, in that it controls for prospective endogeneity between digital financial inclusion and remittance. This outcome further confirms that household remittances have a significant positive impact on the likelihood that household members will at least hold an e-bank account. This result provides additional support to the first hypothesis ($H_1$). Thus, it can be said that higher remittances provide extra cash to households for some time, which raises the demand for bank deposits, as financial establishments are considered safe places for households to keep their cash. These results are highly consistent with those of some recent studies [20].

**Table 6.** Total remittances: instrumental variable.

| | Total Remittance | | |
|---|---|---|---|
| | **The Household Has an e-Bank Account** | **Household Uses Mobile Banking** | **Household Uses an ATM Card** |
| Remittance (Log) | 0.029 *** (0.011) | −0.010 (0.012) | 0.017 * (0.011) |
| The age of the household head | 0.039 *** (0.002) | 0.039 *** (0.004) | 0.017 *** (0.003) |
| Number of household members | 0.022 ** (0.007) | 0.021 (0.008) | 0.001 (0.007) |
| The household head's level of education | −0.001 (0.001) | −0.004 ** (0.003) | −0.007 *** (0.003) |
| Female household head (%) | 0.000 (0.001) | −0.001 * (0.002) | −0.002 * (0.002) |
| Destination: | | | |
| European countries | −0.042 (0.041) | 0.022 (0.0399) | 0.041 (0.029) |
| Middle East | −0.027 (0.0371) | 0.0133 ** (0.044) | 0.037 (0.072) |
| East Asia | 0.007 (0.049) | −0.081 (0.071) | −0.031 (0.069) |
| Other countries | 0.071 (0.081) | 0.051 (0.211) | 0.013 (0.074) |
| Employment status of migrants: | | | |
| Paid employment | −0.231 (0.024) | −0.032 (0.039) | −0.024 (0.039) |
| Small-scale trade | 0.211 ** (0.071) | −0.067 (0.074) | 0.055 (0.066) |
| Student | 0.081 (0.061) | 0.071 (0.218) | 0.029 (0.211) |
| Unemployment | 0.078 (0.071) | −0.127 (0.088) | 0.071 (0.067) |
| Others | 0.013 (0.073) | −0.044 (0.211) | 0.063 (0.101) |
| Gender (Female migrant) | −0.127 *** (0.039) | −0.031 (0.59) | −0.114 ** (0.0510 |
| Const. | −0.112 (0.129) | 0.701 *** (0.155) | 0.398 *** (0.192) |
| Rural | Yes | Yes | Yes |
| Region | Yes | Yes | Yes |
| R-squared | 0.316 | 0.193 | 0.273 |
| KP rk LM statistic (*p*-value) | 62.288 | 43.407 | 42.308 |
| | (0.000) | (0.000) | (0.000) |
| CD Wald F statistic | 32.290 | 23.674 | 22.881 |
| Sargan stat. (*p*-value) | 11.725 | 8.919 | 0.775 |
| | (0.001) | (0.002) | (0.401) |

Note: *, **, and *** represent significance levels at 10%, 5%, and 1%, respectively.

**Table 7.** Internal remittance: instrumental variable.

| | Internal Remittance | | |
|---|---|---|---|
| | The Household Has an e-Bank Account | Household Uses Mobile Banking | Household Uses an ATM Card |
| Remittance (Log) | 0.041 *** (0.011) | −0.012 (0.023) | 0.028 * (0.022) |
| The age of the household head | 0.039 *** (0.002) | 0.019 *** (0.004) | 0.021 *** (0.003) |
| Number of household members | 0.025 ** (0.007) | 0.021 (0.008) | 0.010 (0.007) |
| The household head's level of education | −0.003 (0.001) | −0.004 ** (0.003) | −0.007 *** (0.003) |
| Female household head (%) | 0.011 (0.001) | −0.003 * (0.001) | −0.002 * (0.002) |
| Destination: | | | |
| European countries | −0.032 (0.038) | 0.021 (0.028) | 0.033 (0.041) |
| Middle East | −0.020 (0.033) | 0.127 ** (0.049) | 0.049 (0.072) |
| East Asia | 0.019 (0.047) | −0.092 (0.071) | −0.031 (0.69) |
| Other countries | 0.071 (0.081) | 0.051 (0.219) | 0.027 (0.074) |
| Employment status of migrants: | | | |
| Paid employment | −0.021 (0.024) | −0.031 (0.039) | −0.26 (0.034) |
| Small-scale trade | 0.133 ** (0.071) | −0.066 (0.075) | 0.046 (0.068) |
| Student | 0.081 (0.066) | 0.071 (0.221) | 0.028 (0.113) |
| Unemployment | 0.082 (0.073) | −0.211 (0.088) | 0.062 (0.066) |
| Others | 0.115 (0.073) | −0.434 (0.221) | 0.064 (0.101) |
| Gender (Female migrant) | −0.218 *** (0.039) | −0.014 (0.049) | −0.211 ** (0.039) |
| Const. | −0.118 (0.217) | −0.615 *** (0.159) | 0.399 *** (0.168) |
| Rural | Yes | Yes | Yes |
| Region | Yes | Yes | Yes |
| R-squared | 0.329 | 0.157 | 0.283 |
| KP rk LM statistic (*p*-value) | 61.142 | 42.631 | 41.281 |
| | (0.000) | (0.000) | (0.000) |
| CD Wald F statistic | 34.216 | 24.373 | 23.536 |
| Sargan stat. (*p*-value) | 11.812 | 8.413 | 0.601 |
| | (0.001) | (0.002) | (0.398) |

Note: *, **, and *** represent significance levels at 10%, 5%, and 1%, respectively.

The coefficient of remittance was also found to have a positive relationship with the probability of a household's use of mobile banking for financial transactions. This result supports the second hypothesis (H$_2$). This outcome further reinforces the recent findings using the linear probability model and probit estimates. However, the indicative migrant remittance coefficients point out that incidence of remittances increases the likelihood of a household's use of mobile banking. The coefficient of household remittances is not statistically positively significant when digital financial inclusion is proxied by the use of ATM cards. This result does not support the third hypothesis (H$_3$). The coefficient of remittances is not important when digital financial inclusion is determined by the use of ATM cards. This implies that remittance receipt does not affect the probability that household members use ATM cards. An almost-identical outcome was found in this case when the disaggregated migrant remittances were considered.

Consistent with the research assumptions, the migrant's location, when corresponding to the Middle East, affects the household's digital financial inclusion. That is, migrant household members in the Middle Eastern region were more likely to use mobile banking than were internal migrant households. Additionally, the number of adult members in a household increased the likelihood that a household held an e-bank account. Nevertheless, the increased age of an adult member in a household increased the probability that any household member held an e-bank account, used ATM cards, or used mobile banking. Furthermore, having a student migrant from the household was more probably associated with the household having an e-bank account. Notwithstanding this, that fact that the migrant from the household was female made it less likely that the household was associated with an e-bank account or used mobile banking, compared to situations with male migrants from the households.

**Table 8.** International remittance: instrumental variable results.

| | International Remittance | | |
| --- | --- | --- | --- |
| | **The Household Has an e-Bank Account** | **Household Uses Mobile Banking** | **Household Uses an ATM Card** |
| Remittance (Log) | 0.079 *** (0.031) | −0.004 (0.031) | 0.022 * (0.021) |
| The age of the household head | 0.029 *** (0.003) | 0.032 *** (0.004) | 0.016 *** (0.003) |
| Number of household members | 0.007 (0.007) | 0.010 (0.008) | 0.001 (0.008) |
| The household head's level of education | −0.003 * (0.001) | −0.007 *** (0.003) | −0.007 *** (0.003) |
| Female household head (%) | 0.001 (0.001) | −0.001 (0.001) | −0.003 *(0.001) |
| Destination of migrants: | | | |
| European countries | −0.398 *** (0.139) | 0.039 (0.201) | −0.101 (0.103) |
| Middle East | −0.401 *** (0.104) | 0.144 (0.108) | −0.101 (0.102) |
| East Asia | −0.311 ** (0.100) | −0.621 (0.100) | −0.104 (0.111) |
| Other countries | −0.319 (0.138) | 0.0701 (0.122) | −0.124 (0.122) |
| Employment status of migrants: | | | |
| Paid employment | −0.003 (0.0035) | −0.031 (0.039) | −0.031 (0.038) |
| Small-scale trade | 0.193 ** (0.064) | −0.033 (0.071) | 0.041 (0.071) |
| Student | 0.073 (0.082) | 0.0211 (0.088) | 0.011 (0.077) |
| Unemployment | 0.071 (0.071) | −0.087 (0.075) | 0.041 (0.069) |
| Others | 0.136 (0.011) | −0.010 (0.111) | 0.071 (0.110) |
| Gender (Female migrant) | −0.188 *** (0.072) | −0.030 (0.049) | −0.101 ** (0.049) |
| Const. | 0.149 (0.109) | 0.489 *** (0.211) | 0.491 *** (0.163) |
| Rural | Yes | Yes | Yes |
| Region | Yes | Yes | Yes |
| R-squared | −0.109 | 0.152 | 0.189 |
| KP rk LM statistic (*p*-value) | 18.143 | 18.852 | 18.408 |
| | (0.000) | (0.000) | (0.000) |
| CD Wald F statistic | 11.458 | 9.608 | 9.431 |
| Sargan stat. (*p*-value) | 3.128 | 9.511 | 0.311 |
| | (0.0712) | (0.0012) | (0.627) |

Note: *, **, and *** represent significance levels at 10%, 5%, and 1%, respectively.

Table 9 presents the propensity score matching analysis of the effect of a household's remittance receipt, given total remittances, on digital financial inclusion. However, Table 10 reports the separate data analyses of internal and international remittances, respectively, using three matching algorithms, namely, nearest neighbor, stratification, and kernel matching. The matching algorithms provide immensely corresponding estimates of the impact of migrant household remittances on digital financial inclusion. The contradictory approach reveals that, in sum, remittances significantly increase savings accounts provided, by 11.8–13.7%. These outcomes reveal that migrant remittance-receiving households have a significant positive impact on digital financial inclusion. This result is quite consistent with the conclusions of previous studies on the impact of remittances on digital financial inclusion [20–23].

**Table 9.** Total remittances: propensity score matching results.

| **Variables (Outcome)** | **Matching Algorithm** | **ATET** | **S.E** | ***t*-Test** |
| --- | --- | --- | --- | --- |
| The household has an e-bank account | Nearest Neighbor | 0.137 *** | 0.039 | 2.847 |
| The household has an e-bank account | Kernel | 0.131 *** | 0.029 | 2.369 |
| The household has an e-bank account | Stratification | 0.118 *** | 0.030 | 3.612 |
| Household uses mobile banking | Nearest Neighbor | −0.082 * | 0.049 | −1.623 |
| Household uses mobile banking | Kernel | −0.002 | 0.038 | −0.065 |
| Household uses mobile banking | Stratification | 0.006 | 0.037 | 0.191 |
| Household uses an ATM card | Nearest Neighbor | −0.88 | 0.071 | −1.326 |
| Household uses an ATM card | Kernel | −0.038 | 0.036 | −1.101 |
| Household uses an ATM card | Stratification | −0.049 | 0.049 | −0.881 |

Note: * and *** represent significance levels at 10% and 1%, respectively.

**Table 10.** Internal and international remittances: propensity score matching results.

| Outcome Variables | Matching Algorithm | Remittance | | | | | |
|---|---|---|---|---|---|---|---|
| | | Internal | | | International | | |
| | | ATET | S.E | *t*-Test | ATET | S.E | *t*-Test |
| The household has an e-bank account | Nearest Neighbor | 0.131 *** | 0.048 | 2.361 | −0.002 | 0.041 | −0.075 |
| The household has an e-bank account | Kernel | 0.113 *** | 0.031 | 2.851 | 0.029 | 0.028 | 1.166 |
| The household has an e-bank account | Stratification | 0.118 *** | 0.028 | 2.711 | 0.028 | 0.031 | 1.121 |
| Household uses mobile banking | Nearest Neighbor | −0.082 * | 0.048 | −1.568 | −0.038 | 0.049 | −1.178 |
| Household uses mobile banking | Kernel | −0.002 | 0.039 | −0.071 | −0.073 | 0.049 | −1.378 |
| Household uses mobile banking | Stratification | 0.006 | 0.041 | 0.125 | −0.048 | 0.037 | −1.255 |
| Household uses an ATM card | Nearest Neighbor | −0.081 | 0.070 | −1.445 | −0.069 | 0.055 | −1.226 |
| Household uses an ATM card | Kernel | −0.036 | 0.035 | −1.114 | −0.065 * | 0.039 | −1661 |
| Household uses an ATM card | Stratification | −0.049 | −0.49 | −0.889 | −0.071 | 0.038 | −1.232 |

Note: * and *** represent significance levels at 10% and 1%, respectively.

## 7. Conclusions

This paper seeks to understand the impacts of migrant remittances on the use of means of digital financial inclusion within households in Bangladesh by using a migration and remittance household survey. Analyzing the results shows that the use of e-bank accounts in conventional financial-management banks and mobile banking for receiving remittances and financial transactions among migrant households has increased. The use of ATM cards by households for financial transactions has not been significantly affected. The article explains that remittance-recipient households play an important role in strengthening digital financial inclusion in a country.

The significant positive relation between penetration to financial services and remittances described in this paper may be implemented in the context of state policy decisions. For example, Bangladesh is one of the most remittance-dependent countries in the world, and one which has various practical barriers to receiving remittances. Despite the various costs and expenses associated with managing remittance receipts by remittance-recipient households in Bangladesh, the receipt may increase the likelihood of digital financial inclusion by increasing the flow of remittances sent to the country through the formal channels. Undoubtedly, the findings of this study demonstrate a strong linkage relative to multi-dimensional aspects of inclusion in the field of digital financial inclusion, which will help those in the technological and service sectors to shape future policies.

The findings of the study have theoretical, methodological, and practical contributions. From a methodological and theoretical perspective, previous studies have investigated technology acceptance using TRAM. However, its acceptability has not been investigated at the household level, particularly in the most important financial sectors in developing countries, such as remittances. This study has attempted to empirically explore the acceptance and adoption of digital finance in urban and rural settings.

Despite the contributions of the study, it also has some limitations. The main limitation of this study is the determination of sample frame and size. Thus, the findings obtained from this study may be generalizable only to the study area, and may not be generalizable beyond the area. Therefore, a more in-depth study can be achieved by increasing the sample size, especially the stratified size, as well as by collecting data through social stratification, especially in rural areas. It is hoped that this study will lead to more research in this area.

**Author Contributions:** Conceptualize, K.A.M.; data collection, K.A.M. and K.M.F.; analysis, K.A.M.; methodology, K.A.M.; discussions, K.A.M.; limitations and future direction, K.M.F.; referencing, K.A.M.; overall guidelines and proof reading, K.M.F. All authors have read and agreed to the published version of the manuscript.

**Funding:** This research received no external funding.

**Institutional Review Board Statement:** Not applicable.

**Informed Consent Statement:** Not applicable.

**Data Availability Statement:** Not applicable.

**Conflicts of Interest:** The authors declare no conflict of interest.

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
