# Peer review of "Digital Financial Inclusion and Remittances: An Empirical Study on Bangladeshi Migrant Households"

_fintech, doi:10.3390/fintech2040038_

Round 1

Reviewer 1 Report

Digital Financial Inclusion and Remittances: An Empirical Study on Bangladeshi Migrant Households  

Fintech-2563329

Section: Abstract & Introduction

The Abstract and introduction should be better explained and motivated further.  

Section: Literature Review

The “Review of Literature” section should be improved. In the “Review of Literature” section, the authors should focus on reviewing previous studies that are significantly relevant to the objective of the study and should draw a research gap on the same.

The research gap is missing in the literature review section.

The hypothesis needs to be developed.

The authors need to update the literature review section after using some recently published literature. These include –

-         Does COVID-19 influence in reshaping the banking habits of the individual? An empirical investigation. International Journal of Electronic Finance11(4), 345-363.

Section: Method

·         Study region:- Why the chosen area is so significant for the study?

·         Sampling method & participants:- the sample design needs to be described in more detail.

·         The method used needs to be extended and described in a more detailed way, with proper justifications.

Section: Findings  

The statistical interpretation of the results required analytical discussion.

Section: Conclusion:

This section should be better explained and motivated.

The managerial implications are missing.

Please explain how the findings of this study contribute to theory and practice.

Please discuss study limitations and based on these limitations, proposed future research directions.

Others:

Please pay attention to a few language issues that require correction.

·         Please pay attention to a few language issues that require correction.

Author Response

I have attached the PDF file here which contains the detailed text.

Reviewer 2 Report

Digital financial inclusion is indeed an important issue in emerging economies. Remittance flow through digital financial systems would help countries avoid informal money transfer, too. However, there are many areas where this article needs to improve. My major comments are as follows:

1.       Include policy implications in the abstract.

2.       “The article concludes that remittance flows may enhance access to and use of digital financial inclusion …” how? I found that remittance was significantly related to a bank account, not related to mobile banking (digital), and was weakly related to ATM cards (digital). Please explain.

3.       In the first and second paragraphs of the Introduction, the authors need to discuss financial inclusion through digital financial technologies. Reduction of remittance and its motivations are not major concerns of this study.

4.       There is no theoretical background of the study in the Introduction. Digital financial inclusions, digital financial technologies, etc. have not been introduced. The authors truly mentioned that sending remittance through a formal system increased financial inclusion. They need to show clearly how digital financial inclusions are positively affected by remittance.

5.       Section 2 could be merged with Section 1 to prepare a better introduction and background of the study.

6.       The novelty and contribution of the study are not evident in the Introduction.

7.       Authors did not develop any hypothesis.

8.       Please provide a detailed outline of the survey and database. The process of data collection, sampling process, validity and reliability of measures, data sufficiency, representativeness etc need to be highlighted.

9.       Add a subsection on variable definitions and measurements.

10.   The results are grossly inconsistent. For example, the mean of household using ATM card is only 7.2% (Table 1.1), but when the full sample is divided into receiver of remittance and not (Table 1.2), the mean of household using ATM is 74.6% and 64.4%. How is this possible? Similar inconsistencies are found in other measurements, too.

11.   The paper produces serious confusion on the use of the terms “financial inclusions” and “digital financial inclusions”. What is the objective of the study? financial inclusion or digital financial inclusion. Clarify and make proper revisions throughout the manuscript.

12.   The paper could benefit from a copyediting by a competent editor.

A moderate english revision is needed for clarity of idea and logical flow. Grammatical errors are also found in some places.  

Author Response

(The authors gave the same response as above.)

Round 2

Reviewer 2 Report

I can see a clear effort of the authors to respond to my previous comments. While I am satisfied with their efforts, there are more issues to resolve before this paper can be considered for publication.

1. There is still room for improving the introduction and background section. Make tit concise and relevant to the hypothesis. Avoid unnecessary discussion.

2. "The basic objective of this study is to find the effect of migrant household remittances on digital financial services in Bangladesh. Specifically, to observe what induces the use of digital financial inclusions among Bangladeshi remittance recipient households and whether there is any variation in the use of such digital financial services among non-remittance recipient households." The first part is consistent with the hypothesis. But the second part is not - the study is not about "what induces the use of financial inclusions".

3. Where in the empirical model you reflect "perceived usefulness", "attitude" "perceived ease of use", and "behavioral intention" of the conceptual model"

4. Typo: add H2 in the second hypothesis

5. Data: who conducted the survey? Please be specific. If authors themselves conducted the survey, did they go through the ethical approval process? 

6. Inconsistency in results is still present in the descriptive statistics. For ex. use of mobile banking. Descriptive statistics is an important issue. Please go through the whole calculations again including estimation models to check correctness.

7. Please be more specific in discussing policy implications.

8. Copyediting is highly recommended.

Copyediting is highly recommended.

Round 3

Reviewer 2 Report

It is important to know more about the survey. Who conducted the survey? Please include all relevant information about the survey.
